# Co-Transcriptional Regulation of HBV Replication: RNA Quality Also Matters

**DOI:** 10.3390/v16040615

**Published:** 2024-04-16

**Authors:** Guillaume Giraud, Khadija El Achi, Fabien Zoulim, Barbara Testoni

**Affiliations:** 1INSERM U1052, CNRS UMR5286, Centre de Recherche en Cancérologie de Lyon, Université Claude Bernard Lyon 1, 69008 Lyon, Francefabien.zoulim@inserm.fr (F.Z.); 2The Lyon Hepatology Institute EVEREST, 69003 Lyon, France; 3Hospices Civils de Lyon, Hôpital Croix Rousse, Service d’Hépato-Gastroentérologie, 69004 Lyon, France

**Keywords:** HBV, cccDNA, transcription, epigenetics, RNA splicing, RNA methylation, RNA polyadenylation

## Abstract

Chronic hepatitis B (CHB) virus infection is a major public health burden and the leading cause of hepatocellular carcinoma. Despite the efficacy of current treatments, hepatitis B virus (HBV) cannot be fully eradicated due to the persistence of its minichromosome, or covalently closed circular DNA (cccDNA). The HBV community is investing large human and financial resources to develop new therapeutic strategies that either silence or ideally degrade cccDNA, to cure HBV completely or functionally. cccDNA transcription is considered to be the key step for HBV replication. Transcription not only influences the levels of viral RNA produced, but also directly impacts their quality, generating multiple variants. Growing evidence advocates for the role of the co-transcriptional regulation of HBV RNAs during CHB and viral replication, paving the way for the development of novel therapies targeting these processes. This review focuses on the mechanisms controlling the different co-transcriptional processes that HBV RNAs undergo, and their contribution to both viral replication and HBV-induced liver pathogenesis.

## 1. Introduction

About 300 million people worldwide are chronically infected with the hepatitis B virus (HBV). Chronic hepatitis B (CHB) increases the risk of developing severe liver diseases and is the main driver of hepatocellular carcinoma (HCC), responsible for around 800,000 deaths/year. Current treatments based on nucleoside analogues (NUCs) are effective at keeping the infection under control and are well-tolerated by patients [1]. Nevertheless, these treatments do not fully eradicate HBV owing to their inability to target the viral minichromosome (known as covalently closed circular DNA (cccDNA)), and thus require life-long administration. There is therefore an urgent need to develop new therapeutic strategies that directly target cccDNA to cure HBV completely or functionally [2].

Several strategies targeting cccDNA are currently envisaged [3]. First, to impair cccDNA formation, treatments could target host proteins belonging to the DNA repair system, although targeting such proteins would have dramatic consequences on cell homeostasis. Second, to target the established cccDNA molecule, ongoing pre-clinical studies using gene-editing approaches (e.g., CRISPR/Cas9 and base editing) have shown promising results at degrading and/or silencing cccDNA [4,5,6,7]. Nevertheless, these approaches raise many concerns regarding off-target effects, which can be deleterious for cell homeostasis, and delivery [8]. Cytokines such as interferon alpha (IFNα) can also be used to induce cccDNA degradation or silencing [9], although these strategies need to be optimized to be more efficient and better tolerated by patients.

Much effort, therefore, still has to be made before achieving complete loss of cccDNA. Given the efficacy of FDA-approved drugs targeting epigenetic factors in the treatment of other cancers, targeting the transcriptional activity of cccDNA may constitute an attractive strategy to cure HBV. Moreover, growing evidence supports a role for HBV RNA co-transcriptional processing in CHB development, paving the way for novel therapeutic targets. Generating knowledge on cccDNA transcriptional and post-transcriptional regulation is thus critical to identify new therapeutic targets against CHB.

This review presents the current knowledge on molecular mechanisms controlling cccDNA transcriptional activity and particularly focuses on the co-transcriptional regulation of HBV RNAs.

## 2. cccDNA Transcription: A Key Step of HBV Replication

Several reviews have extensively highlighted the different mechanisms involved in controlling HBV gene expression at the transcriptional level [10]. Hence, we provide only a brief overview of the subject.

### 2.1. Generalities on cccDNA

cccDNA is considered to be the key molecule of HBV replication. It is formed through a complex process dependent on the host DNA repair machinery using relaxed circular DNA imported into the nucleus after HBV entry into the hepatocyte [11]. During its formation, cccDNA is chromatinized and adopts a stable episomal structure, which is the unique template for the transcription of the main six viral mRNAs by the host RNA polymerase II (RNAP II) [12,13,14]. This includes pre-genomic RNA (pgRNA), which is retro-transcribed by the host viral polymerase while being encapsidated to form new infectious particles or to replenish the pool of nuclear cccDNA. The transcription of cccDNA is thus a critical step to ensure optimal HBV replication.

### 2.2. cis-Element Controlling cccDNA Transcriptional Activity

Four RNAP II promoters located at different positions of the cccDNA genome initiate the transcription of the six viral mRNAs [15]. The *Core* promoter generates the two longer-than-genome mRNAs, namely the pgRNA and the precore mRNA (3.5 kb RNA). These two mRNAs initiate their synthesis at different locations [15,16]. In vitro studies suggested that the transcription of these two viral mRNAs is initiated at two overlapping and independent promoters that are differentially regulated by host factors [17]. However, this observation has not been confirmed in vivo and still requires further investigation. The SPI promoter generates PreS1 (2.4 kb RNA) mRNA, while SPII produces PreS2 and S (both 2.1 kb) mRNAs [18,19,20,21]. Recent studies indicate that, as the *Core* promoter, SPII promoter supports the initiation of the preS2 and S mRNAs at different locations [13,14]. Finally, the X promoter transcribes the different X mRNAs, which also have different initiation sites [22,23,24].

The activity of these promoters is strengthened by two transcriptional enhancers, the EnhI and the EnhII regions located upstream of the *X* and *Core* promoters, respectively [25,26,27,28,29]. In vitro studies suggested that these two enhancers ensure that the above transcripts are expressed at the right time. While the EnhI region seems to be involved in the transcription of early HBV transcripts (X), the EnhII region allows the transcription of late HBV transcripts. This differential activity of the two enhancers seems to depend on host factors that are recruited to these two regulatory regions [30].

Secondary structures have been identified in the abovementioned regulatory sequences. Biswas et al. identified a G-quadruplex (G4) motif in the SPII promoter of the HBV genotype B [31]. This dynamic structure is recurrently observed in regulatory sequences of mammalian and viral genomes and actively contributes to gene expression regulation [32]. In vitro studies demonstrated that this G-quadruplex promotes the activity of the SPII promoter. Whether this regulation is active in a replicative model, as well as its underlying molecular mechanism, remain unknown [31]. Another G4 structure has been identified in the *Core* promoter. Mutations of this structure were shown to result in an increased level of HBe antigen (HBeAg), the secretion of HBsAg antigens, and a decreased level of intracellular HBcAg, suggesting that this G4 regulates HBV replication [33]. However, this study did not functionally link this effect to a potential contribution of this G4 to the activity of the *Core* promoter. A recent study performed a cartography of G4 structures present on the HBV genome and demonstrated the functional role of two of those, located in the EnhI region, in cccDNA transcription by promoting its phase separation in the nucleus of infected primary hepatocytes via the G4-dependent recruitment of the fused in sarcoma (FUS) protein [34].

### 2.3. Trans-Factors Controlling cccDNA Transcriptional Activity

HBx is the most potent activator of cccDNA transcription, as well as the most extensively described one. HBx-defective viruses are unable to synthesize HBV RNAs due to their closed chromatin, promoted by host restriction factors [35,36]. HBx is thought to act primarily by degrading the SMC5/6 complex in a DNA damage-binding protein 1 (DDB1) ubiquitin ligase-dependent manner, and thus prevents its recruitment to cccDNA [37,38,39]. Other mechanisms have also been proposed and thoroughly described [40]. Mutations in HBc were reported to result in a decreased level of HBV RNAs, suggesting that HBc is also an activator of cccDNA transcription. The authors showed that HBc is recruited to HBV cccDNA alongside acetyltransferases, such as the CREB-binding protein (CBP), thus creating an opened chromatin environment adapted to an optimized transcription [41]. However, the precise role of HBc in cccDNA regulation remains elusive [42].

Aside from viral factors, several host transcription factors and cofactors were described as regulators of cccDNA transcription. Among these, the hepatocyte transcription factors hepatocyte nuclear factor 1 and 4a (HNF1a and HNF4a) are critical activators of HBV transcription, partly explaining the hepato-tropism of HBV [43,44]. The identification of such transcription factors as regulators of HBV transcription has paved the way for the development of drugs targeting these proteins. This is, for example, the case of baicalin, which was shown to inhibit viral RNA transcription by preventing the dimerization of HNF4a, and to increase the effect of NUC treatment in HepG2.2.15 cells [45]. Interestingly, HNF4a is a target of the metabolic sensor, peroxisome proliferator-activated receptor gamma coactivator 1-alpha (PGC-1a). Starvation activates HBV gene expression via the PGC-1a/HNF4a axis, demonstrating that the metabolic status of the infected organism also plays a critical role in HBV transcription [46]. Other transcription factors activated by PGC-1a, such as forkhead box protein O1 (FOXO1), are also implicated in the activation of HBV transcription [47].

The host genome organizer CCCTC-binding factor (CTCF) was recently shown to repress HBV transcription by binding to two CTCF sites located in the EnhI region [48]. Moreover, CTCF binding within this region was reported to regulate the specific phasing of nucleosomes to maintain an open chromatin conformation and regulate HBV transcription [49]. The identification of CTCF as a regulator of cccDNA nucleosome positioning and transcription suggests that cccDNA may adopt a tri-dimensional conformation depending on its transcriptional activity. Nevertheless, addressing the tri-dimensional organization of cccDNA is currently technically challenging due to the small size of this episome.

Non-coding RNAs (ncRNAs) represent an additional layer of HBV transcriptional regulation by modulating the activity of cccDNA transcription factors. The ncRNA HOX transcript antisense RNA (*HOTAIR*), highly induced by HBV infection, was shown to promote HBV transcription by increasing SP1 recruitment to cccDNA [50]. Whether *HOTAIR* or other ncRNAs interact directly with cccDNA has never been shown.

A number of additional regulators of cccDNA transcription have been identified and a non-exhaustive list is provided in Figure 1 [51,52,53,54,55].

### 2.4. Epigenetic Modulation of cccDNA Transcriptional Activity

As for the human genome, cccDNA-bound histones undergo post-translational modifications (PTMs) that are functionally linked to cccDNA transcriptional activity. The acetylation status of histones is a major driver of cccDNA transcription and is regulated by histone acetyltransferases and deacetylases; for example, the histone deacetylase 11 (HDAC11) deacetylates histone H3 lysine 9 (H3K9) and 27 (H3K27) residues triggering the epigenetic silencing of cccDNA transcription and the subsequent restriction of HBV replication in Huh7 cells [56]. In contrast, the histone acetyltransferase 1 (HAT1) acetylates several residues of histones H3 (K27) and H4 (K5, K12), promoting cccDNA transcription and HBV replication in several in vitro and in vivo models [57]. Other PTMs of cccDNA-bound histones have been identified and regulate cccDNA transcriptional activity. Indeed, succinylation of H3K122 catalyzed by lysine acetyltransferase 2A (KAT2A) and removed by sirtuin 7 (SIRT7) is associated with transcriptional activation [58,59]. Conversely, trimethylation of H3K9 or H3K27 catalyzed by SET domain bifurcated histone lysine methyltransferase 1 (SETDB1) and the polycomb repressive complex 2 (PRC2), respectively, are associated with transcriptional silencing of cccDNA [54,60,61]. Similarly, methylation of the arginine residue of histone H4 (H4R3me) catalyzed by PRMTs is associated with the repression of cccDNA transcriptional activity [53,62]. These data thus highlight a tight epigenetic regulation of cccDNA activity.

The viral protein HBx orchestrates different molecular mechanisms to reshape cccDNA chromatin in favor of an active chromatin [35,63]. A number of chromatin-modifying enzymes were shown to interact with HBx proteins in several models. For instance, acetyltransferase p300 is recruited to cccDNA via HBx to acetylate the histone H3 [64,65]. Recently, the interaction between HBx and Spindlin1 was shown to promote cccDNA transcription by allowing a chromatin switch from a repressive H3K9me3 to an active H3K4me3 conformation [66]. HBx also counteracts the recruitment of HBV restriction factors, such as HMGB1 or SETDB1, to cccDNA, thus preventing its epigenetic silencing [60,67]. Finally, HBx represses the expression of the suppressor of zeste 12 homolog (SUZ12) subunit of the PRC2 complex, thus impeding the deposition of H3K27me3 repressive marks [54].

Interestingly, current treatments for CHB are associated with the rewiring of cccDNA chromatin. Indeed, the interferon alpha treatment of HepG2-NTCP and HepaRG cells is associated with the desuccinylation of cccDNA and a lower deposition of H3K27 acetylation (H3K27ac) contributing to epigenetic silencing [68]. Similarly, long-term telbivudine treatment of CHB patients is associated with the increased deposition of repressive H3K9me3 and H3K27me3 histone marks and with the concomitant reduced deposition of active H3K27ac and H3K56ac marks. These changes are associated with lower levels of 3.5 kb RNAs while cccDNA levels remain stable, strongly suggesting that long-term telbivudine treatment is associated with the silencing of cccDNA transcriptional activity through an epigenetic mechanism [69].

### 2.5. Compartmentalization of cccDNA in the Nucleus of Infected Hepatocytes

Hi-C experiments in primary human hepatocytes (PHHs) demonstrated that cccDNA mainly interacts with open chromatin regions containing CpG islands in a CXXC finger protein 1 (CPF1)-dependent manner. Silencing of this factor decreased the transcriptional activity of cccDNA and the interaction with these particular regions, indicating the importance of these regions in the regulation of HBV transcription [70]. In contrast, the transcriptionally inactive HBx-deficient cccDNA is more frequently located within a heterochromatin hub in the chromosome 19 in HepDE19 cells. The re-expression of HBx induces a positional change towards opened chromatin regions in a structural maintenance of chromosomes protein 5/6 (SMC5/6)-dependent manner [71]. These studies thus demonstrate that cccDNA is not randomly localized in the nucleus of infected hepatocytes and that it interacts with specific host genome regions depending on its transcriptional activity (recently reviewed in [72]). Interestingly, cccDNA is able to undergo liquid-to-liquid phase separation (LLPS) [34]. LLPS results from the local concentration of proteins and nucleic acids dedicated to a specific biological process, thus creating membrane-less organelles that improve the efficacy of all steps involved in gene expression [73]. cccDNA phase separation occurs in FUS-containing nuclear speckles located in euchromatin regions in the nucleus of infected hepatocytes and is dependent on G4-mediated FUS binding to cccDNA. It is therefore tempting to speculate that HBV is able to hijack the physiological FUS capacity to bind G4s and promote LLPS, thus ensuring an optimal nuclear environment for the transcription of its genome [34].

**Figure 1 viruses-16-00615-f001:**
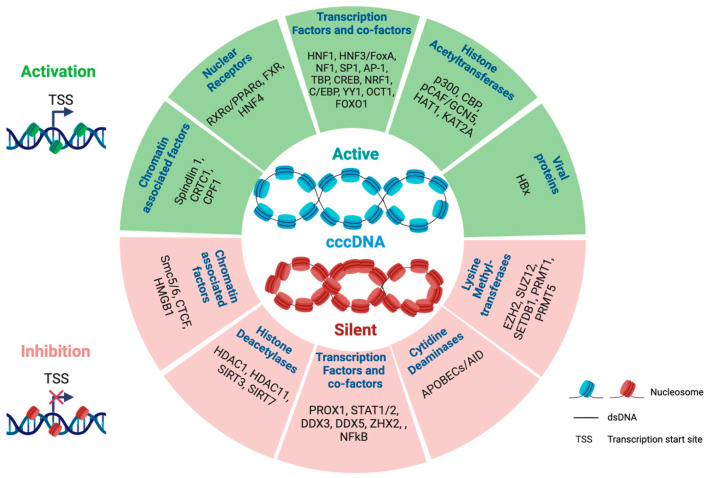
Regulators of HBV cccDNA transcription. Examples of host or viral factors activating (green) or silencing (red) cccDNA transcriptional activity associated with an active (green nucleosomes) or a silent (red nucleosomes) chromatin [37,38,39,40,43,47,48,49,50,52,53,54,56,57,58,59,60,62,63,65,66,67,68,70,74,75,76,77]. The image was created using Biorender.com.

## 3. Co-Transcriptional Regulation of HBV RNAs

Transcription not only influences the level of synthesized RNA, but also impacts their quality. Thus, multiple RNA variants can be generated with different roles in HBV replication. This section provides an overview of the current knowledge on co-transcriptional HBV RNA processing and its importance in HBV replication and CHB.

### 3.1. HBV RNA Splicing

HBV RNA splicing is by far the most extensively described co-transcriptional process regulating HBV RNA metabolism. This process has previously been reviewed in several journals reflecting its high importance for HBV biology [78,79,80,81]. HBV spliced transcripts were first observed three decades ago in the serum of CHB patients by Northern blot analysis. The authors used a probe against the Core region and identified a transcript of 2.2 kb [82]. Su et al. confirmed that this 2.2 kb transcript corresponds to a splice variant (SV), most likely derived from the pgRNA, by in vitro experiments in Huh-7 hepatoma cells [83]. Moreover, in vitro experiments performed in COS-7 and in Huh-7 cells identified a spliced variant derived from PreS2/S mRNA [84,85]. To date, 22 HBV SVs have been identified in the serum of patients and in cellular models, and can be generated from all HBV genotypes. These SVs use a unique combination of splice donor and acceptor sites but a single spliced junction can be common to several HBV SVs, making their specific identification difficult [78].

HBV SVs are not able to generate fully functional polymerase or surface proteins. Thus, optimal HBV replication requires a finely tuned regulation of HBV RNA splicing. Several cis-elements and host trans-factors have been identified as key regulators of HBV RNA splicing. These include the post-transcriptional regulatory element (PRE) region, which was highlighted as being critical for the HBV RNAs’ nuclear export [86,87]. Indeed, deletion of the PRE region completely abolished the generation of the SP1 HBV SVs in Huh-7 cells, demonstrating the importance of this regulatory element in this process. Heise et al. showed, in vitro, that this effect is mediated by an exon splicing enhancer contained within this region [86]. As expected, these regulatory elements recruit splicing factors such as PSF or SRSF2 [86,88]. Interestingly, the overexpression of the HDAC5 histone deacetylase in pCMV-HBV-transfected HEK293T cells was suggested to increase HBV RNA splicing [89]. Although the authors did not correlate the splicing outcome with cccDNA chromatin rewiring, these data indicate that cccDNA chromatin and HBV RNA splicing are also connected. Further investigations are nonetheless required to validate this hypothesis.

Contrary to other viruses such as HIV-1, HBV RNA splicing is not essential for HBV replication. HBV SVs can be retro-transcribed and generate genome-defective HBV particles, which are thought to be associated with liver diseases [90]. Moreover, several studies showed a higher level of HBV SVs one to three years prior to the development of HCC and a correlation with HBV viral load [91]. This observation strongly suggests that HBV SVs contribute to the pathogenicity and/or the persistence of HBV. Several studies highlighted the cross-talk between HBV SVs and the host immune system, which plays important roles in the pathogenesis induced by HBV. In particular, Chen et al. reported a negative correlation between HBV SVs and the interferon-alpha (IFN-α) response, suggesting that HBV SVs could participate to the innate immune escape of HBV during CHB [79]. Strikingly, the overexpression of the HB spliced protein (HBSP) or reverse transcriptase’-RNaseH (RT’-RH) proteins in hepatoma cells, similarly to POL, prevented the nuclear translocation of STAT1 and STAT2 and the activation of STAT1 in response to IFN-α treatment [79,92,93]. These two non-canonical HBV proteins are encoded by SP1, and are major HBV splice variants and contain a truncated POL RT domain and a complete RNAse H domain of the viral polymerase. In addition to the POL TP domain, the RNAse H domain was demonstrated to impair the IFN-α signaling pathway [92]. Moreover, HBSP was also shown to induce a HBSP-specific T-cell response in mice and in patients. This response is thought to contribute to liver damage induced by HBV through cytokines, such as IFN-γ, produced by T cells [94].

HBV SV-encoded proteins were also reported to regulate the level of HBV replication acting as self-restriction factors. Indeed, the HpZ/P’ protein, encoded by the HBV SVs bearing the major spliced junctions between the 5′ splice site at position 2450 and the 3′ splice site at 489, was shown to counteract the effect of suppressor of var1, 3-like 1 (SUPV3L1) on cccDNA transcription and to hamper the acetylation of histones H3 and H4 on HBV minichromosome [88].

### 3.2. HBV RNA Methylation

Methylation of the N6-Methyladenosine (m^6^A) is the most frequent RNA modification and occurs co-transcriptionally. This modification plays a major role in RNA metabolism and was shown to regulate the replication of several viruses such as HIV-1 [95,96]. Strikingly, HBV RNAs are methylated in HepG2 cells, and MeRIP-Seq experiments precisely mapped the adenine residue at position 1907 (genotype D) (Figure 2). This residue is located at the basis of the ε stem-loop and is found twice in the pgRNA and preCore RNA, but only once in the other sub-genomic RNAs [97]. This modification is deposited by a methyltransferase complex composed of methyltransferase like 3 and 14 (METTL3 and METTL14), the recruitment of which on HBV RNAs is dependent on HBx in HepG2-NTCP cells and in PHHs [98]. Accordingly, RNAs generated from a HBx-null pHBV plasmid are less methylated compared to those generated from the wild-type one. The recruitment of METTL3 and METTL14 to HBV RNAs and the subsequent methylation of HBV RNAs cannot be restored by a transcriptionally inactive HBx mutant, strongly indicating that, as for host transcripts, the methylation of HBV RNA depends on transcription [98].

Interestingly, this particular residue is located at a previously identified G4 in HBV RNAs [99,100]. Recently, G4 structures were shown to be critical for the recruitment of METTL14, which is essential for the methyl-transferase activity of METTL3 [101]. Whether this is the case for the HBV G4 is not yet known.

Mutagenesis experiments showed a dual function of HBV RNA methylation depending on whether it impacts the 5′ (only found in the pgRNA and pC RNA) or the 3′ ε-loop (found in all HBV RNAs). Indeed, the methylation of HBV RNAs at the 5′ loop positively regulates the reverse transcription of the pgRNA while the methylation at the 3′ loop impairs HBV RNA stability [97]. A possible explanation for the role of the m^6^A modification of the 5′ ε-loop stems from the observation that it favors the interaction with the core protein and the subsequent encapsidation of the pgRNA. In agreement with these data, methylated pgRNAs are more present in Core particles compared to unmethylated ones in Huh-7 cells [102]. m^6^A-modified HBV RNAs recruit fragile X mental retardation protein (FMRP) and YTH domain containing 1 (YTHDC1) m^6^A readers, which are important for nuclear export of HBV transcripts. Accordingly, repression of these factors or mutation of the m^6^A residue at both ε-loops lead to the nuclear accumulation of HBV transcripts, highlighting an extra role for m^6^A modifications in the HBV life cycle [103].

Methylation of HBV RNA was also identified as a critical regulator of the innate immune response. Indeed, mutagenesis experiments similar to those described above in HepG2 cells demonstrated that the HBV RNA m^6^A methylation impairs their recognition by the RIG-I protein and the subsequent phosphorylation of interferon regulatory factor 3 (IRF3) [104]. These data show that m^6^A methylation not only plays a major role in the regulation of the HBV life cycle but also participates in the pathogenesis induced by HBV.

More recently, a novel m^6^A residue was identified by MeRIP at position 1616 of the coding sequence of HBx. Mutagenesis experiments and depletion of YTHDF2 m^6^A readers showed that this methylation represses HBx expression at both RNA and protein levels. Interestingly, the expression of HBs is also affected by this modification located at the 3′ UTR of the S mRNA. Whether this is a consequence of HBx repression or a more direct effect on the S transcript itself is unknown and requires further investigation [105].

Altogether, these data highlight the plethora of viral functions of the m^6^A modification of HBV RNAs.

### 3.3. HBV RNA Polyadenylation

HBV RNAs, as in the case of most RNAP II-generated transcripts, are polyadenylated at their 3′ end [106]. Polyadenylation is tightly linked to transcription termination and is initiated by the recognition of a polyA signal (PAS), which is generally an AAUAAA motif on RNA. In vitro studies identified a non-canonical PAS, TATAAA, deviating from the AAUAAA motif by only one base [107]. Strikingly, this motif is not recognized during the first passage of the RNAP II generating the precore and the pgRNA, which are thus longer than the genome. A 3.9 kb long HBx RNA not terminating at this motif has also been identified, though it is less frequent than the canonical 0.7 kb long HBx RNA [108]. This raises the question of how this motif is recognized, knowing that the surface mRNAs mostly use this motif as a termination signal. Some explanation came from studies of the ground squirrel hepatitis B virus (GSHBV). Mutagenesis experiments of sequences upstream of the TSS of the pgRNA allowed the identification of three regions termed PS1, 2, and 3, which are important for the proper usage of the HBV PAS. Whether these elements act at the DNA or RNA level was not determined by these studies, but their efficient recognition relies on the passage of RNAP II [109,110]. So far, these sequences have not been identified in the human HBV and the mechanisms underlying the recognition of this HBV PAS remain unclear.

Fractionation experiments in Huh-7 and HepG2 cells transfected with HBV DNA showed a nuclear accumulation of the long HBx transcript, which does not end at the PAS motif in the first passage of the RNAP II, indicating a link between transcription termination and HBx RNA export [108]. The mechanism underlying the poor shuttling of the long HBx RNA remains obscure, considering that it contains twice the PRE region responsible for HBV RNA export.

Recently, we proposed a model where the DEAD-box helicases DDX5 and DDX17 were responsible for inhibiting PAS recognition, resulting in longer HBV transcript 3′UTR RNA destabilization and decreased HBx protein levels [111]. These data highlight the importance of transcriptional fidelity in finely tuning HBV replication.

Interestingly, the sequence of this HBV PAS corresponds to a TATA box that is normally located at the level of promoters [112]. In vitro studies demonstrated that this motif allowed the recruitment of general transcription factors (GTFs) and the TATA-box binding protein (TBP), and displays promoter activity in luciferase reporter assays. This element was shown to be essential for HBV replication, as the restoration of the canonical PAS sequence completely impaired the production of viral DNA intermediates [112]. These observations thus highlighted a dual function for HBV PAS. At the DNA level, it acts as a TATA box element, while at the RNA level, it acts as a termination signal for RNAP II. In eukaryotes, chromatin looping events between promoters and terminators have been identified. These chromatin loops create a roadblock for RNAP II, causing a pause in the activity of the polymerase and allowing the efficient recognition of the PAS. In addition to this role, this particular genome topology facilitates the recycling of RNAP II and accelerates reinitiation of the transcription cycle [113,114,115]. If cccDNA were to adopt such a topology, it would place this HBV PAS/TATA in close proximity with the HBV promoters that do not contain canonical TATA boxes, thus contributing to their transcriptional activity. Due to the small size of cccDNA, testing this hypothesis remains challenging.

The nucleotidic composition of the poly (A) tail is a combination of A and G residues. This mixed tailing is ensured by a complex composed of zinc finger CCHC-type containing 14 (ZCCHC14), PAP associated domain containing 5 and 7 (PAPD5 and PAPD7) proteins, which are responsible for the incorporation of G residues. These G residues prevent the recognition of the poly-A tail by poly-A ribonucleases, thus stabilizing RNA. In line with this, the repression of ZCCHC14 or dual repression of PAPD5 and PAPD7 in several hepatic cellular models resulted in the destabilization of HBV RNAs. Further studies identified the recruitment of this complex to the PRE SLα region, indicating a direct effect of these proteins [116,117]. Strikingly, this complex is targeted by therapeutic compounds such as AB452 or RG7834, which repress HBV replication and are considered as potential therapeutic targets against CHB [117].

### 3.4. Capping of HBV RNAs

Transcripts generated by RNAP II are capped at their 5′ end. This cap mainly consists in a 7-methyl-guanosine that is linked through a 5′ to 5′ triphosphate bridge to the first transcribed nucleotide, which is methylated on the ribose O-2 position. The deposition of this cap, which is dependent on three enzymatic activities, occurs during transcription in a process dependent on RNAP II, highlighting its co-transcriptional nature. This cap structure has multiple roles in mRNA metabolism by recruiting protein complexes such as eukaryotic translation initiation factor 4E (EIF4E) [118,119]. Although it was not formerly demonstrated, indirect evidence based on the use of techniques specific to capped RNA showed that HBV RNAs are capped [13,14].

As mentioned above, the pgRNA contains two ε stem loop structures, among which only the 5′ ε is required for encapsidation [120]. Transfection of Huh-7 cells with plasmids that synthesize pgRNA with the 5′ ε motif at different distances of the 5′ end highlighted a position effect of this particular motif on encapsidation. Further in vitro studies indicated that the 5′ cap is essential for pgRNA encapsidation and suggested that POL requires both the 5′ ε motif and the 5′ cap to be recruited on pgRNA [121]. Co-immunoprecipitation experiments in HEK 293T cells revealed the interaction between HBV POL and the cap-binding factor eif4E, which is dependent on the 5′ ε motif but independent of the 5′ cap. Interestingly, eif4E is encapsidated together with POL and pgRNA, and the authors proposed that it could be important for the synthesis of the positive strand, but further investigations are required to corroborate this hypothesis [122]. Whether the interaction between POL and eif4E is required for the position effect of the 5′ ε has not been addressed in this study but could explain the cap-dependence of the encapsidation process.

## 4. Conclusions

The HBV community is investing heavily in finding a way to either degrade or silence cccDNA in order to obtain a functional, or ideally complete, cure for this disease. Due mainly to safety reasons, there are still quite a few hurdles to clear before reaching this crucial goal. Transcription not only influences the level of RNAs but also directly impacts their metabolism. This review highlighted the different co-transcriptional processes that regulate the metabolism of HBV RNAs and presented their importance in both HBV replication and HBV-induced liver disease. Identifying the molecular mechanisms that control such processes could thus offer an alternative approach to design new therapeutic drugs that impair viral replication and slow down disease progression.

## 5. Future Directions

Drugs targeting co-transcriptional processes already exist and have shown promising results in pre-clinical studies. This is, for instance, the case for METTL3 inhibitors in acute myeloid leukemia [123]. Interestingly, RG7834 targets PAPD5 and PAPD7, which maintain the integrity of the poly-A tail and was demonstrated to destabilize HBV RNAs in cellular models [117]. This drug was tested alone and in combination with entecavir or IFNα in woodchuck and mice that were chronically infected with HBV, and displayed promising effects on decreasing viral load, suggesting that it could be used as a cure for CHB [124,125]. Altogether, these observations offer interesting perspectives for the identification of host targets to improve current CHB treatment strategies.

## Figures and Tables

**Figure 2 viruses-16-00615-f002:**
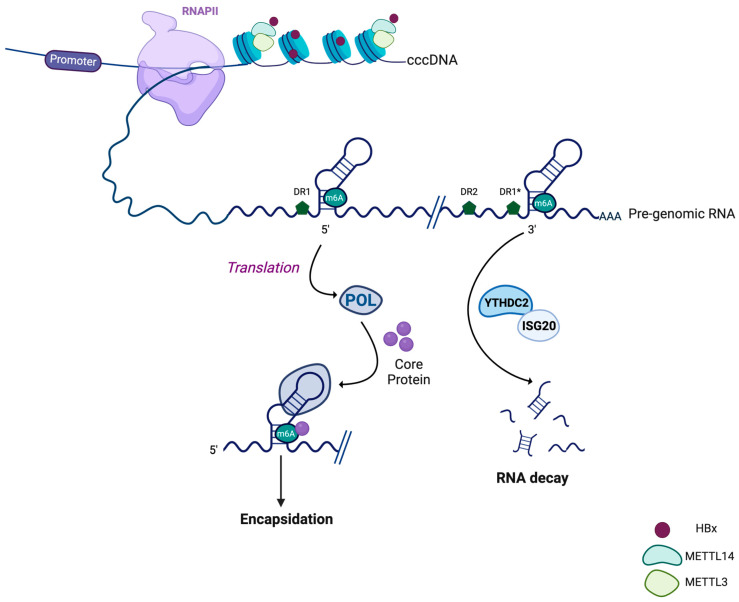
Dual role of m6A methylation of HBV RNAs. HBV RNAs are co-transcriptionally methylated at position 1907 corresponding to the basis of the ε-loop. The methylation is ensured by the two methyltransferases METTL3 and METTL14, which are recruited during HBV transcription by the HBx viral protein (top part). This important secondary structure is found twice in pgRNA but only once in sub-genomic RNAs. Depending on the localization of the ε-loop, m^6^A methylation of HBV RNAs has different effects. Indeed, modification of the residue of the 5′ structure allows its recognition by the viral Core protein and its subsequent encapsidation and reverse transcription by the viral polymerase, thus initiating a new cycle of viral replication. In contrast, m^6^A methylation at the 3′ ε-loop allows its recognition by the methylation reader YTHDC2, which recruits the ISG20 exonuclease and impairs HBV RNA stability. DR, direct repeat; RNAPII, RNA polymerase II; ISG20, interferon stimulated gene 20; YTHDC2, YTH domain containing 2; METTL, Methyltransferase-like. The figure was created using Biorender.com.

## Data Availability

Not applicable.

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
