# Peer review of "Co-Transcriptional Regulation of HBV Replication: RNA Quality Also Matters"

_viruses, 2024, doi:10.3390/v16040615_

Round 1

Reviewer 1 Report

Comments and Suggestions for Authors

Chronic hepatitis B virus (HBV) infection poses a significant global public health threat, contributing to substantial morbidity and mortality. Nucleotide or nucleoside analogs have significantly improved the quality of life for patients with chronic hepatitis B. However, achieving a functional cure remains challenging due to the persistent presence of covalently closed circular DNA (cccDNA) in infected hepatocytes. In their review, Giraud et al. highlighted the current understanding of the molecular mechanisms controlling cccDNA transcriptional activity and the co-transcriptional regulation of HBV RNAs. This review provides invaluable insights for researchers and clinicians in the field of HBV. While the paper is well-written and suitable for publication, several points should be modified.

Specific comments:

1.       All factors depicted in Fig. 1 should be referenced. Additionally, if factors described in the text are involved in the regulation of HBV cccDNA transcription, they should all be included in Fig. 1 if applicable.

2.       Line 180, "H3K27m3e" should be corrected to "H3K27me3."

3.       Line 188, similar to "H3K27ac," "H3K56Ac" should be corrected to "H3K56ac."

4.       Line 224, although this reference was published in 1989, using "Lately" may not be appropriate.

Comments on the Quality of English Language

Since there are some expressions in the text that may be unclear, please review and refine the overall manuscript a bit.

Author Response

We thank Reviewer 1 for the positive appreciation of our manuscript.

Point-by-point response to specific comments:

All factors depicted in Fig. 1 should be referenced. Additionally, if factors described in the text are involved in the regulation of HBV cccDNA transcription, they should all be included in Fig. 1 if applicable.

Authors' reply: As suggested by the reviewer, we slightly re-organized the figure to accommodate factors that were described in the text and not included in the original version. References corresponding to the proteins included in each group were added in brackets. Four additional references were added to the list (122-125) so that every protein appearing in the figure is referenced.

  • Line 180, "H3K27m3e" should be corrected to "H3K27me3."
  • Line 188, similar to "H3K27ac," "H3K56Ac" should be corrected to "H3K56ac."
  • Line 224, although this reference was published in 1989, using "Lately" may not be appropriate.

Authors' reply: All the modifications suggested by the reviewer have been incorporated in the revised version of the manuscript and highlighted in yellow.

Since there are some expressions in the text that may be unclear, please review and refine the overall manuscript a bit.

Authors' reply: The revised version of the manuscript underwent English proof-reading  (highlighted in red) and we hope that is now better readable and suitable to be published in Viruses.

Reviewer 2 Report

Comments and Suggestions for Authors

This review paper summarizes recent topics related to covalently closed circular DNA (cccDNA). Based on the authors' research experience regarding cccDNA, it is well-organized, with appropriate diagrams. However, many abbreviations are listed without their full spelling. Therefore, it is highly recommended to use an English proofreader with scientific knowledge.

Abstract

Lane 12: Chronic Hepatitis B (CHB)

Lane 13: “HBV” should change to “HB virus (HBV)”

Main text

Lane 46: “IFNα” should change to “interferon α (IFNα)”

Lane 66: rcDNA delete (There was no rcDNA in other parts)

Lane101: “HBeAg” should change to “HBe Antigen (HBeAg)”

Lane 107: “FUS” should change to “fused ins sarcoma (FUS)”.

Lane 112: “DDB1” should change to “DNA damage-binding protein 1”.

Lane 116: “CBP” should change to “CREB binding protein”.

Lane 122: “HNF1” should change to “hepatocyte nuclear factor 1 (HNF1)”.

Lane 128: “PGC-1a” should change to “peroxisome proliferator-activated receptor gamma coactivator 1-alpha (PGC-1a)”.

Lane 131: “FOXO1” should change to “forkhead box protein O1”.

Lane 133: “CTCTF” should change to “CCCTC-binding factor (CTCF)”.

Lane 144: “HOTAIR” should change to “HOX transcript antisense RNA (HOTAIR)”.

Figure 1

“Viral proteins” should change to “Viral Proteins”.

“Cytidine Aminase” should change to “Cytidine Deaminase”.

“Transcription Factors and co-factors” should change to “Transcription Factors and Co-factors”.

“Inhibition” should change to “Repression”.

“dsDNA” should change to “double stranded DNA”.

Lane 157: For example, the histone deacetylase HDAC11 deacetylates lysine 9 (H3K9) and 27 (H3K27)

For example, histone deacetylase 11 (HDAC11) deacetylates histone H3 lysine 9 (H3K9) and 27 (H3K27)

Lane 160: “HAT1” should change to “histone acetyltransferase 1 (HAT1)”.

Lane 164: “KAT2A” should change to “lysine acetyltransferase 2A (KAT2A)”.

Lane 164: “SIRT7” should change to “sirtuin 7 (SIRT7)”.

Lane 166: “H3K27me3” should change to “H3K27 trimethylation (H3K27me3)”.

Lane 166: “SETDB1” should change to “SET domain bifurcated histone lysine methyltransferase 1 (SETDB1)”.

Lane 178: “HMGB1” should change to “high mobility group box 1 (HMGB1)”.

Lane 179: “SUZ12” should change to “suppressor of zeste 12 homolog”.

Lane 180: “PRC2” should change to “polycomb repressive complex 2 (PRC2)”.

Lane 183: “NTCP” should change to “Na+/taurocholate cotransporting polypeptide (NTCP)”.

Lane 185: “H3K27ac” should change to “H3K27 acetylation (H3K27ac)”.

Lane 188: “H3K56Ac” should change to “H3K56ac”.

Lane 194: “CPF1” should change to “CXXC finger protein 1 (CFP1)”.

Lane 200: “SMC5” should change to “Structural maintenance of chromosomes protein 5 (SMC5)”.

Lane 257: “HBSP” should change to “HB spliced protein (HBSP)”.

Lane 257: “RT’-RH” should change to reverse transcriptase’-RNaseH (RT’-RH)”.

Lane 258: “STAT1” should change to “Signal transducer and activator of transcription 1 (STAT1)”.

Lane 260: “POL” should change to “HBV polymerase (POL)”.

Lane 261: “RNAse H” should change to “RH”.

Lane 269: “SUPV3L1” should change to “suppressor of var1, 3-like 1”.

Lane 271: Figure 2 delete. Lane 276, (genotype D) (Figure 2).

Lane 272: “m6A” should change to “N6-Methyladnosine (m6A)”.

Lane 279: “METTL3” should change to “methyltransferase like 3 (METTL3)”.

Figure 2

“RNA pol” should change to “RNA polymerase II or RNAPII”.

“DR” should write to “direct repeat”, once.

“Pol” should change to “POL”.

Lane 297: “G-quadruplex (G4)” should change to “G4”.

Lane 309: “FMRP” should change to “fragile X mental retardation protein”.

Lane 309: “YTHD1” should change to “YTH domain containing 1 (YTHD1)”.

Lane 317: “IRF3” should change to “interferon regulatory factor 3 (IRF3)”.

Lane 353: “GTFs” should change to “general transcription factors (GTFs)”.

Lane 353: “TBP” should change to “TATA-box binding protein”.

Lane 374: “ZCCH14” should change to “zinc finger CCHC-type containing 14 (ZCCH14)”.

Lane 374: “PAPD5” should change to “PAP associated domain containing 5 (PAPD5)”.

Lane 390: “eif4e” should change to “eukaryotic translation initiation factor 4E (EIF4E)”.

Comments on the Quality of English Language

I strongly recommend using English editing.

Author Response

We thank Reviewer 2 for the positive evaluation of our manuscript.

Point-by-point response to specific comments:

Abstract

Lane 12: Chronic Hepatitis B (CHB)

Lane 13: “HBV” should change to “HB virus (HBV)”

Main text

Lane 46: “IFNα” should change to “interferon α (IFNα)”

Lane 66: rcDNA delete (There was no rcDNA in other parts)

Lane101: “HBeAg” should change to “HBe Antigen (HBeAg)”

Lane 107: “FUS” should change to “fused ins sarcoma (FUS)”.

Lane 112: “DDB1” should change to “DNA damage-binding protein 1”.

Lane 116: “CBP” should change to “CREB binding protein”.

Lane 122: “HNF1” should change to “hepatocyte nuclear factor 1 (HNF1)”.

Lane 128: “PGC-1a” should change to “peroxisome proliferator-activated receptor gamma coactivator 1-alpha (PGC-1a)”.

Lane 131: “FOXO1” should change to “forkhead box protein O1”.

Lane 133: “CTCTF” should change to “CCCTC-binding factor (CTCF)”.

Lane 144: “HOTAIR” should change to “HOX transcript antisense RNA (HOTAIR)”.

Figure 1

“Viral proteins” should change to “Viral Proteins”.

“Cytidine Aminase” should change to “Cytidine Deaminase”.

“Transcription Factors and co-factors” should change to “Transcription Factors and Co-factors”.

“Inhibition” should change to “Repression”.

“dsDNA” should change to “double stranded DNA”.

Lane 157: For example, the histone deacetylase HDAC11 deacetylates lysine 9 (H3K9) and 27 (H3K27)

For example, histone deacetylase 11 (HDAC11) deacetylates histone H3 lysine 9 (H3K9) and 27 (H3K27)

Lane 160: “HAT1” should change to “histone acetyltransferase 1 (HAT1)”.

Lane 164: “KAT2A” should change to “lysine acetyltransferase 2A (KAT2A)”.

Lane 164: “SIRT7” should change to “sirtuin 7 (SIRT7)”.

Lane 166: “H3K27me3” should change to “H3K27 trimethylation (H3K27me3)”.

Lane 166: “SETDB1” should change to “SET domain bifurcated histone lysine methyltransferase 1 (SETDB1)”.

Lane 178: “HMGB1” should change to “high mobility group box 1 (HMGB1)”.

Lane 179: “SUZ12” should change to “suppressor of zeste 12 homolog”.

Lane 180: “PRC2” should change to “polycomb repressive complex 2 (PRC2)”.

Lane 183: “NTCP” should change to “Na+/taurocholate cotransporting polypeptide (NTCP)”.

Lane 185: “H3K27ac” should change to “H3K27 acetylation (H3K27ac)”.

Lane 188: “H3K56Ac” should change to “H3K56ac”.

Lane 194: “CPF1” should change to “CXXC finger protein 1 (CFP1)”.

Lane 200: “SMC5” should change to “Structural maintenance of chromosomes protein 5 (SMC5)”.

Lane 257: “HBSP” should change to “HB spliced protein (HBSP)”.

Lane 257: “RT’-RH” should change to reverse transcriptase’-RNaseH (RT’-RH)”.

Lane 258: “STAT1” should change to “Signal transducer and activator of transcription 1 (STAT1)”.

Lane 260: “POL” should change to “HBV polymerase (POL)”.

Lane 261: “RNAse H” should change to “RH”.

Lane 269: “SUPV3L1” should change to “suppressor of var1, 3-like 1”.

Lane 271: Figure 2 delete. Lane 276, (genotype D) (Figure 2).

Lane 272: “m6A” should change to “N6-Methyladnosine (m6A)”.

Lane 279: “METTL3” should change to “methyltransferase like 3 (METTL3)”.

Figure 2

“RNA pol” should change to “RNA polymerase II or RNAPII”.

“DR” should write to “direct repeat”, once.

“Pol” should change to “POL”.

Lane 297: “G-quadruplex (G4)” should change to “G4”.

Lane 309: “FMRP” should change to “fragile X mental retardation protein”.

Lane 309: “YTHD1” should change to “YTH domain containing 1 (YTHD1)”.

Lane 317: “IRF3” should change to “interferon regulatory factor 3 (IRF3)”.

Lane 353: “GTFs” should change to “general transcription factors (GTFs)”.

Lane 353: “TBP” should change to “TATA-box binding protein”.

Lane 374: “ZCCH14” should change to “zinc finger CCHC-type containing 14 (ZCCH14)”.

Lane 374: “PAPD5” should change to “PAP associated domain containing 5 (PAPD5)”.

Lane 390: “eif4e” should change to “eukaryotic translation initiation factor 4E (EIF4E)”.

Authors' reply: All the suggested modifications in the text and figures have been incorporated in the revised version of the manuscript and highlighted in yellow.

I strongly recommend using English editing.

Authors' reply: The revised version of the manuscript underwent English proofreading (highlighted in red) and we now hope that it is better readable and suitable for publication in Viruses.

Round 2

Reviewer 2 Report

Comments and Suggestions for Authors

All reviewer requests have been resolved.